# Identification of CFAP52 as a novel diagnostic target of male infertility with defects of sperm head-tail connection and flagella development

Hui-Juan Jin[1†], Tiechao Ruan[2†], Siyu Dai[3†], Xin-Yan Geng[1], Yihong Yang[4,5*], Ying Shen[3*], Su-Ren Chen[1*]

[1]Key Laboratory of Cell Proliferation and Regulation Biology, Ministry of Education, Department of Biology, College of Life Sciences, Beijing Normal University, Beijing, China; [2]Department of Pediatrics, West China Second University Hospital, Sichuan University, Chengdu, China; [3]Key Laboratory of Obstetrics and Gynecologic and Pediatric Diseases and Birth Defects of the Ministry of Education, Sichuan University, Chengdu, China; [4]Reproduction Medical Center of West China Second University Hospital, Key Laboratory of Obstetric, Gynecologic and Pediatric Diseases and Birth Defects of Ministry of Education, Sichuan University, Chengdu, China; [5]NHC Key Laboratory of Chronobiology, Sichuan University, Chengdu, China

**\*For correspondence:**
yyhpumc@foxmail.com (YY);
yingcaishen01@163.com (YS);
chensr@bnu.edu.cn (SRC)

[†]These authors contributed equally to this work

**Competing interest:** The authors declare that no competing interests exist.

**Abstract** Male infertility is a worldwide population health concern. Asthenoteratozoospermia is a common cause of male infertility, but its etiology remains incompletely understood. No evidence indicates the relevance of *CFAP52* mutations to human male infertility. Our whole-exome sequencing identified compound heterozygous mutations in *CFAP52* recessively cosegregating with male infertility status in a non-consanguineous Chinese family. Spermatozoa of *CFAP52*-mutant patient mainly exhibited abnormal head-tail connection and deformed flagella. *Cfap52*-knockout mice resembled the human infertile phenotype, showing a mixed acephalic spermatozoa syndrome (ASS) and multiple morphological abnormalities of the sperm flagella (MMAF) phenotype. The ultra-structural analyses further revealed a failure of connecting piece formation and a serious disorder of '9+2' axoneme structure. CFAP52 interacts with a head-tail coupling regulator SPATA6 and is essential for its stability. Expression of microtubule inner proteins and radial spoke proteins were reduced after the CFAP52 deficiency. Moreover, CFAP52-associated male infertility in humans and mice could be overcome by intracytoplasmic sperm injection (ICSI). The study reveals a prominent role for CFAP52 in sperm development, suggesting that CFAP52 might be a novel diagnostic target for male infertility with defects of sperm head-tail connection and flagella development

## eLife assessment

This study provides **useful** information on the function of a ciliary and flagellar-associated protein, CFAP52, in the assembly of sperm head-tail connecting apparatus (HTCA) and tail formation in humans and mice. The significance is to identify CFAP52 as a genetic factor for asthenoteratozo-ospermia with a mixed acephalic spermatozoa syndrome (ASS) and multiple morphological abnor-malities of the sperm flagella (MMAF) phenotype. The strength of the study is that the experimental evidence using CFAP52 loss-of-function in mice is **solid** to support that CFAP52 is essential for sperm motility and male fertility by contributing to HTCA and 9+2 axoneme, corroborating the sperm phenotypes of human patients with compound heterozygous mutations in CFAP52.

## Introduction

Spermiogenesis, a final phase of spermatogenesis, involves the development of several important organelles and cellular structures, including the acrosome, flagella, and head-tail connecting piece. Disruption of spermiogenesis will lead to male infertility, which is a global health challenge affecting more than 6% of the males (*Chen et al., 2016*). Male infertility can be clinically diagnosed as azoospermia, oligozoospermia, asthenozoospermia, or teratozoospermia (*Jiao et al., 2021*). However, only ~4% infertile men obtain a clear genetic diagnosis and most of the genetic pathogenic causes have not been explored (*Tüttelmann et al., 2018*).

Multiple morphological abnormalities of the sperm flagella (MMAF) and acephalic spermatozoa syndrome (ASS) are two common forms of asthenoteratozoospermia. MMAF usually results from defects in axoneme-based flagellar development. With the development of whole-exome sequencing (WES) technology, DNAH1, CFAP43, CFAP44, CFAP69, CFAP251, ARMC2, TTC21A, TTC29, CFAP58, CFAP91, DNAH8, CFAP47, DNAH10, DNAH17, and WDR63 *Ben Khelifa et al., 2014*; *Coutton et al., 2019*; *Dong et al., 2018*; *He et al., 2020*; *Kherraf et al., 2018*; *Liu et al., 2019*; *Liu et al., 2020a*; *Liu et al., 2021*; *Lu et al., 2021a*; *Martinez et al., 2020*; *Tang et al., 2017*; *Tu et al., 2021*; *Wang et al., 2019*; *Zhang et al., 2020* have been suggested as MMAF-associated genes. Our groups have reported mutations in CFAP65, QRICH2, CEP78, CEP128, CFAP70, and AKAP3 in Chinese patients with MMAF (*Jin et al., 2023*; *Shen et al., 2019*; *Zhang et al., 2019*; *Zhang et al., 2022a*; *Zhang et al., 2022b*; *Yang et al., 2020*). However, the etiology of MMAF remains incompletely understood.

ASS is characterized by decapitated flagella and tailless sperm heads in semen and ultimately causes male sterility. Knockout animal evidence suggest that CNTROB, SPATA6, SUN5, PMFBP1, SPATC1L, FAM46C, and CNTLN are critical for the sperm-tail integrity (*Kim et al., 2018*; *Liska et al., 2009*; *Shang et al., 2017*; *Yuan et al., 2015*; *Zhu et al., 2018*; *Zheng et al., 2019*; *Zhang et al., 2021a*). However, in the clinic, only pathogenic mutations in SUN5 and PMFBP1 (*Elkhatib et al., 2017*; *Shang et al., 2017*; *Zhang et al., 2021b*; *Zhu et al., 2016 Liu et al., 2020b*; *Lu et al., 2021b*; *Sha et al., 2019*) have been identified in a large cohort of ASS patients. ASS usually results from defects in the formation of the head-tail coupling apparatus (HTCA) in the sperm neck region. Ultrastructural studies have reveal that HTCA has a structure lining the implantation fossa of the nucleus called the basal plate (Bp) and a region conforming to the concavity of the Bp called the capitulum (Cp). Extending backwards from the Cp are nine cylindrically segmented columns (Sc). The proximal centriole (Pc) is also a structural element of HTCA, and it is enclosed in a cylindrical niche beneath the Cp (*Dooher and Bennett, 1973*; *Fawcett and Phillips, 1969*; *Zamboni and Stefanini, 1971*). Although the structure of HTCA has been well described, its molecular compositions, assembly properties, and developmental mechanisms are unknown.

CFAP52 (cilia and flagella associated protein 52, also named WDR16) is a member of the large WD40-repeat protein family. FAP52 and FAP45 are orthologues of CFAP52 and CFAP45 in Chlamydomonas, and a lack of these proteins causes an instability of microtubules (*Owa et al., 2019*). Detachment of the B-tubule from the A-tubule and shortened flagella are observed in Chlamydomonas when both FAP52 and FAP20 are absent (*Owa et al., 2019*). The wdr16/cfap52 gene knockdown in zebrafish causes hydrocephalus (*Hirschner et al., 2007*). In humans, CFAP52 mutations have been identified in individuals who presented situs inversus totalis (*Dougherty et al., 2020*; *TaShma et al., 2015*). However, the physiological roles of CFAP52 in mammals are unknown, and whether CFAP52 variants are associated with human disorders of spermatozoa flagella is still unclear.

In this study, we identified deleterious variants of *CFAP52* in a Chinese infertile family, of which the proband exhibited mixed ASS and MMAF phenotype. By generating *Cfap52*-KO mice, we revealed that loss of CFAP52 led to male sterility. Abnormalities in head-tail connections and flagellar formation were identified in sperm of *Cfap52*-KO mice, photocopying the characteristics of the *CFAP52*-mutant patient. CFAP52 interacts with SPATA6 (a well-known structural protein of the Sc and the Cp of the HTCA) to regulate its expression, involving the formation of an intact HTCA structure. Furthermore, the expression of components of the microtubule inner proteins and the radial spokes were reduced in sperm from *Cfap52*-KO mice and the *CFAP52*-mutant patient, explaining the observed MMAF phenomena at the molecular level. Remarkably, CFAP52-associated male infertility in humans and mice could be overcome by using intracytoplasmic sperm injections (ICSI). Our study suggests that mutations of *CFAP52* can be used as an inherited pathogenic factor and a genetic diagnostic indicator for infertility males with asthenoteratozoospermia.

**Table 1.** Semen analysis in the patient.

| Semen parameters | Patient | Reference limits* |
|---|---|---|
| Sperm volume (ml) | 5.5 | ≥ 1.5 |
| Sperm concentration ($10^6$ / ml) | 11.0 | ≥ 15 |
| Motility (A+B, %) | 6.0 | ≥ 40 |
| Vitality (%) | 68.0 | ≥ 58 |
| Normal spermatozoa (%) | 0.5 | ≥ 4 |
| Defective spermatozoa (%) | 99.5 | - |
| Defective sperm flagella (%) | 50.7 | |
| Short flagella (%) | 6.0 | - |
| Coiled flagella (%) | 30.0 | - |
| Absent flagella (%) | 10.5 | - |
| Sperm decapitation (%) | 34.0 | - |

*Reference limits according to the WHO standards.

# Results

## Identification of *CFAP52* variants in a male infertile family

An infertile man from a nonconsanguineous family was investigated in our research. We analysed the semen parameters of the patient, which mainly manifested as reduced sperm motility and abnormal sperm morphology (*Table 1*). Papanicolaou staining and scanning electron microscopy (SEM) were further used to analyze aberrant morphology of the patient's spermatozoa. Compared to a normal control, many patient's spermatozoa showed multiple morphological abnormalities of the sperm flagella (MMAF), including short, coiled, absent, and irregular-calibre flagella (*Figure 1A, B*). Intriguingly, a high percentage of sperm decapitation was identified in the patient's semen (*Figure 1A, B*). To further explore the changes of sperm ultrastructure in the patient, we performed transmission electron microscopy (TEM). Despite the presence of basal

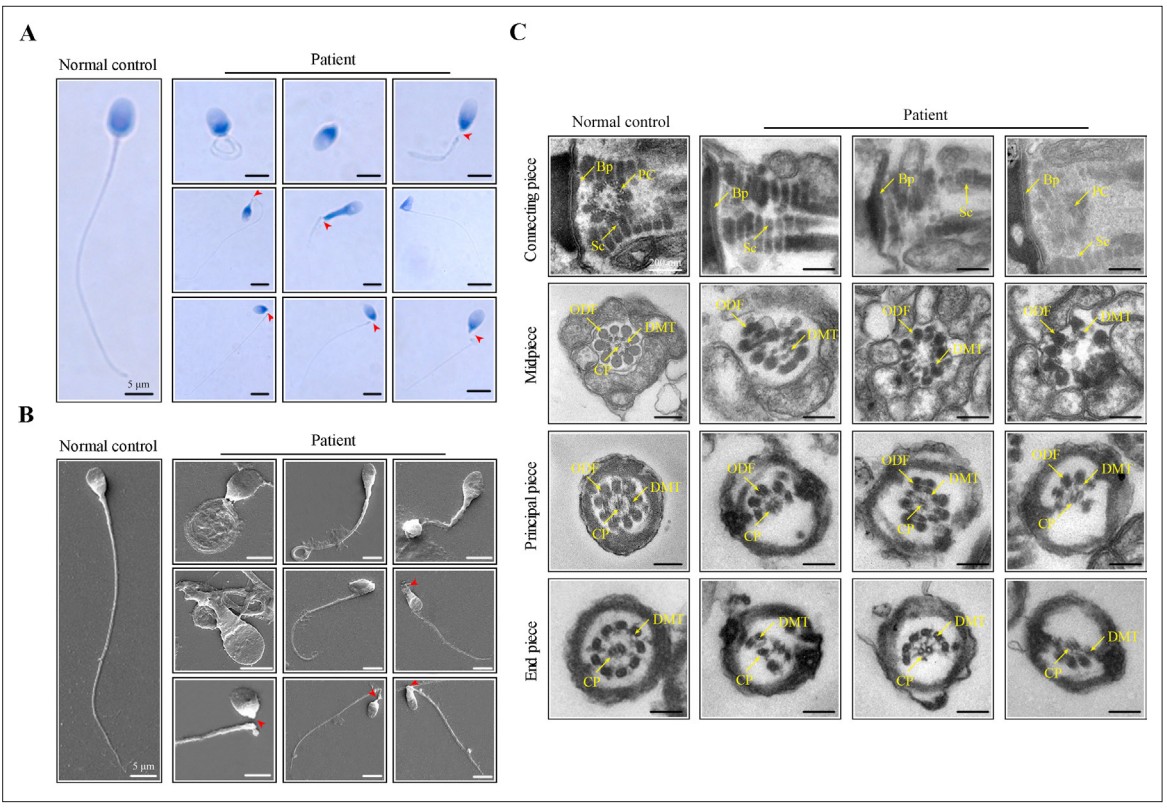

**Figure 1.** Morphological and ultrastructural defects in spermatozoa from the infertile patient. (**A, B**) Papanicolaou staining and scanning electron microscopy (SEM) results showed aberrant spermatozoa morphologies in the infertile patient. Spermatozoa from the patient displayed defective sperm flagella, including absent, short, coiled, and bent flagella. Notably, sperm decapitation was also observed. The red arrowheads indicate the sperm-head connecting piece. Scale bars, 5 µm. (**C**) Transmission electron microscopy (TEM) analyses of the spermatozoa from a healthy control and the infertile patient. Deformed sperm heads, defects of the connecting piece (missing or defective PC and Sc), and abnormalities of the flagella (disorganized arrangements and/or absence of CPs, DMTs, and ODFs) were observed in the patient's spermatozoa. PC, proximal centriole; Sc, segmented column; Bp, basal plate; CP, central-pair microtubule; ODF, outer dense fibre; DMT, doublet microtubules. Scale bars, 100 nm. All experiments were repeated three times with similar results.

**Table 2.** Variant analysis in the patient.

| | | M1 | M2 |
|---|---|---|---|
| | cDNA mutation* | c.203G>T | c.1128G>A |
| | Predicted protein changes | p.S68I | p.W376* |
| | Validate protein changes | p.D24Vfs*5 | p.W376* |
| | Mutation type | Splicing | Nonsense |
| Variants | Genotype | Heterozygous | Heterozygous |
| | ExAC Browser | 0 | 0 |
| | GnomAD | 0 | 0.00000397709 |
| Allele frequency | 1000 Genomes Project | 0 | 0 |
| | dpsi_zscore† | −2.706 | −3.021 |
| Function prediction | SpliceAI score‡ | 0.86 | - |

M1 refers to mutation 1; M2 refers to mutation 2.
*NCBI reference sequence number of *CFAP52* is NM_001080556.2 (https://www.ncbi.nlm.nih.gov/genbank/).
†Absolute values of the score >2 are considered to be deleterious.
‡Scores >0.5 are suggested to affect splicing.

plate (Bp), proximal centrioles (PCs) were incomplete or invisible and some segmented columns (Sc) were defective in the connecting piece of the patient's spermatozoa (*Figure 1C*). Furthermore, disorganized or missing central pairs of microtubules (CPs) were observed in the flagellar axoneme of the patient. Outer dense fibers (ODFs) and doublet microtubules (DMTs) were also partially lacking or disordered (*Figure 1C*).

Whole-exome sequencing was next carried out to explore the underlying genetic cause of asthenoteratozoospermia in the patient. A total of 148 variants *Figure 2—source data 1* were screened out following these criteria: (1) a minor allele frequency was <1% in gnomAD; (2) variants affected coding exons or splice sites; and (3) variants were not predicted benign or likely benign by ACMG. Among these 148 variants, 55 variants were involved in diseases other than male infertility recorded by OMIM. For the remaining 93 variants, we found that only five genes were homozygous, double heterozygous, or X-linked variants, *CFAP52*, *ADAMTS7*, *IRAK1*, *COL4A2*, and *CBFA2T3*. Based on a comprehensive literature review, we found that those genes, with the exception of testis-enriched *CFAP52*, do not exhibit any discernible association with male reproduction.

The missense variant c.203G>T (p.S68I) and nonsense variant c.1128G>A (p.W376*) in *CFAP52* had an extremely low frequency in public databases, including ExAC Browser, gnomAD, and the 1000 genome Project (*Table 2*). Sanger sequencing of all the available family members confirmed that the compound heterozygous *CFAP52* variants in the patient were recessively inherited from their heterozygous parents (*Figure 2A, B*). Notably, the affected sites of the variants were highly conserved among various species, including human, mouse, and rat (*Figure 2C*).

## The *CFAP52* variants leading to absence of its protein expression

To explore the effect of the nonsense mutation (c.1128G>A/p.W376*) on CFAP52 expression, we transfected HEK293T cells with FLAG-tagged WT-*CFAP52* or FLAG-tagged *CFAP52*^c.1128G>A plasmids. Western blotting analysis showed that CFAP52 expression in the cells transfected with the *CFAP52*^c.1128G>A plasmid was not detectable (*Figure 3A*). The c.203G>T variant was predicted to affect the splice site by the SPIDEX and SpliceAI tools (*Table 2*). We performed a mini-gene splicing assay to examine the effect of the c.203G>T variant on *CFAP52* splicing. The electrophoresis results showed that the RT–PCR product of the variant (~263 bp) was shorter than the WT fragment (~400 bp) (*Figure 3B*). Sanger sequencing further indicated that the splicing variant resulted in a deletion of exon 2 in the *CFAP52* transcript (*Figure 3C, D*), which generated a frameshift variant p.D24Vfs*5 of CFAP52. Western blotting results showed that HEK293T cells transfected with the *CFAP52*^c.203G>T plasmid did not express CFAP52 protein (*Figure 3E*). By immunofluorescence staining of spermatozoa, we found that CFAP52 was localized at both HTCA and full-length flagella from the normal

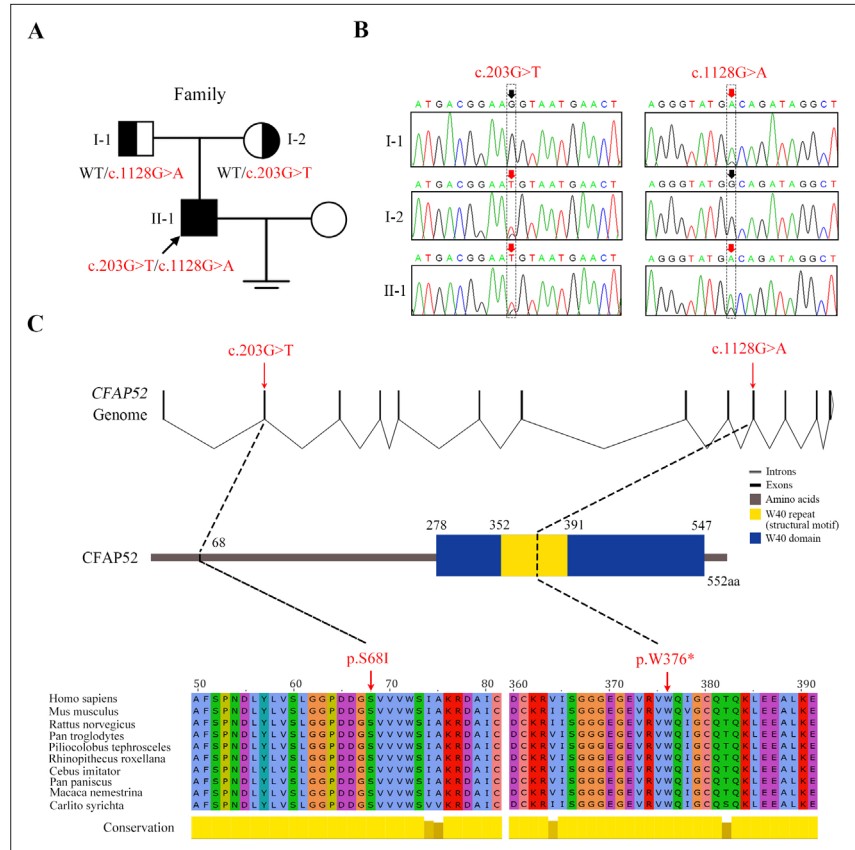

**Figure 2.** Novel mutations in *CFAP52* identified in the infertile family. (**A**) Pedigree of the infertile family; the black arrow indicates the proband (II-1). (**B**) Sanger sequencing confirmed that the proband carried biallelic mutations (c.203G>T/c.1128G>A) in *CFAP52* and that the parents harboured the heterozygous mutation. The red arrows indicate the positions of variants. (**C**) The localization of *CFAP52* variants in the genome, CFAP52 protein structure, and conservation of mutant amino acids in various species. The NCBI reference sequence number of *CFAP52* is NM_001080556.2.

The online version of this article includes the following source data for figure 2:

**Source data 1.** All candidate variants identified in the patients by WES.

control; in contrast, CFAP52 signals were barely detected in the patient's spermatozoa (*Figure 3F*). These mutations in *CFAP52* cause aberrant protein expression and may be the reason for sperm defects in the patient.

### *Cfap52*-KO mice exhibit hydrocephalus and male infertility

Mouse CFAP52 was highly expressed in testes and localized to the manchette (contributing to the formation of the HTCA and the sperm tail) of spermatids and the full-length sperm flagellum (including the HTCA; *Tapia Contreras and Hoyer-Fender, 2020*). To explore the physiological roles of CFAP52, we constructed *Cfap52*-knockout (KO) mice by applying CRISPR/Cas9 technology that targeted exons 2 and 3 of the *Cfap52-201* transcript (ENSMUST00000021287.11; *Figure 4A*). The sequence and location of sgRNAs as well as Sanger sequencing of the founder mice are described (*Figure 4— figure supplement 1*). Male *Cfap52*+/- mice were crossed with female *Cfap52*+/- mice to generate *Cfap52*-KO (*Cfap52*-/-) mice, which were confirmed by genotyping PCR (*Figure 4B*). The CFAP52 protein was completely absent in the *Cfap52*-KO testes compared with the WT testes (*Figure 4C*). *Cfap52*-KO mice suffered from hydrocephalus; the ependymal cilia was sparse under SEM observation and disrupted axonemal structures were identified by TEM analysis (*Figure 4—figure supplement 2*). However, no obvious abnormalities of tracheal cilia were identified by SEM and TEM analyses (*Figure 4—figure supplement 2*). Next, we performed a two-month fertility test of *Cfap52*-KO mice.

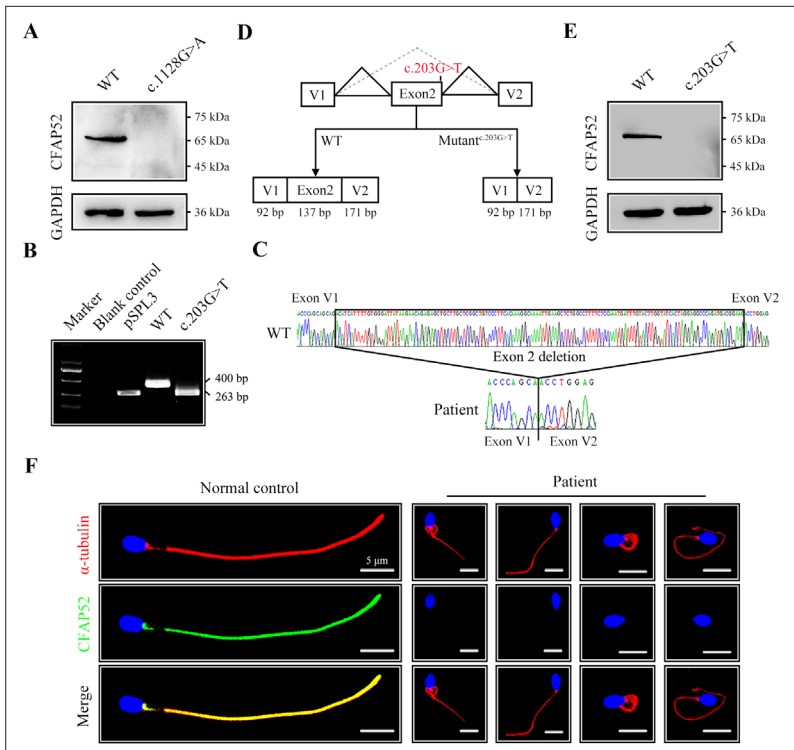

**Figure 3.** The impact of mutations on CFAP52 expression. (**A**) Western blotting analysis showed that CFAP52 expression was not detected in HEK293T cells transfected with the mutant-*CFAP52*$^{c.1128G>A}$ plasmid. (**B**) Electrophoresis showed a decrease in the molecular weight of the PCR products generated from mutant *CFAP52*$^{c.203G>T}$ (263 bp) compared with WT *CFAP52* (400 bp) in the minigene experiment. (**C**) Sanger sequencing of cDNA of the splicing mutation showed the deletion of exon 2 in the c.203G>T variant clone. (**D**) The pattern diagram depicts the adverse effects caused by the *CFAP52* splicing mutation c.203G>T. (**E**) Western blotting analysis showed that HEK293T cells transfected with the mutant *CFAP52*$^{c.203G>T}$ plasmid did not express CFAP52. (**F**) Immunofluorescence staining showed that CFAP52 expression in the patient's spermatozoa was absent compared with that of a healthy control (blue, DAPI; green, CFAP52; red, α-tubulin). Scale bars, 5 μm. All experiments were repeated three times with similar results.

The online version of this article includes the following source data for figure 3:

**Source data 1.** Primers for Sanger sequencing and Minigene.

Although *Cfap52*-KO male mice were able to mate normally with females, they were completely sterile because no pups were born using *Cfap52*-KO males (**Figure 4D, E**).

## *Cfap52*-KO mice produce spermatozoa with abnormal development in head-tail connections and flagella

The testis size, testis/body weight ratio, and serum levels of hormones (e.g. GnRH and testosterone) of *Cfap52*-KO mice was all similar with those of WT mice (**Figure 4—figure supplement 3**). The histological examination of testis sections by periodic acid-schiff staining indicated regular cycles of seminiferous epithelium but flagellar formation defects at stage V-VIII in *Cfap52*-KO mice (**Figure 4—figure supplement 4**). Next, we performed sperm analysis of *Cfap52*-KO mice. Compared with spermatozoa from WT mice, spermatozoa from *Cfap52*-KO mice were not only small in quantity but also had poor motility (**Figure 4F, G**). Papanicolaou staining of spermatozoa indicated that *Cfap52*-KO mice produced headless spermatozoa as well as spermatozoa with abnormal head-tail connections and aberrant tails (**Figure 4H**). Quantitative analyses showed that the decapitated spermatozoa, abnormal head-tail connecting spermatozoa, and spermatozoa with deformed flagella accounted for approximately 40%, 25%, and 30% of the total spermatozoa in *Cfap52*-KO mice, respectively (**Figure 4I**).

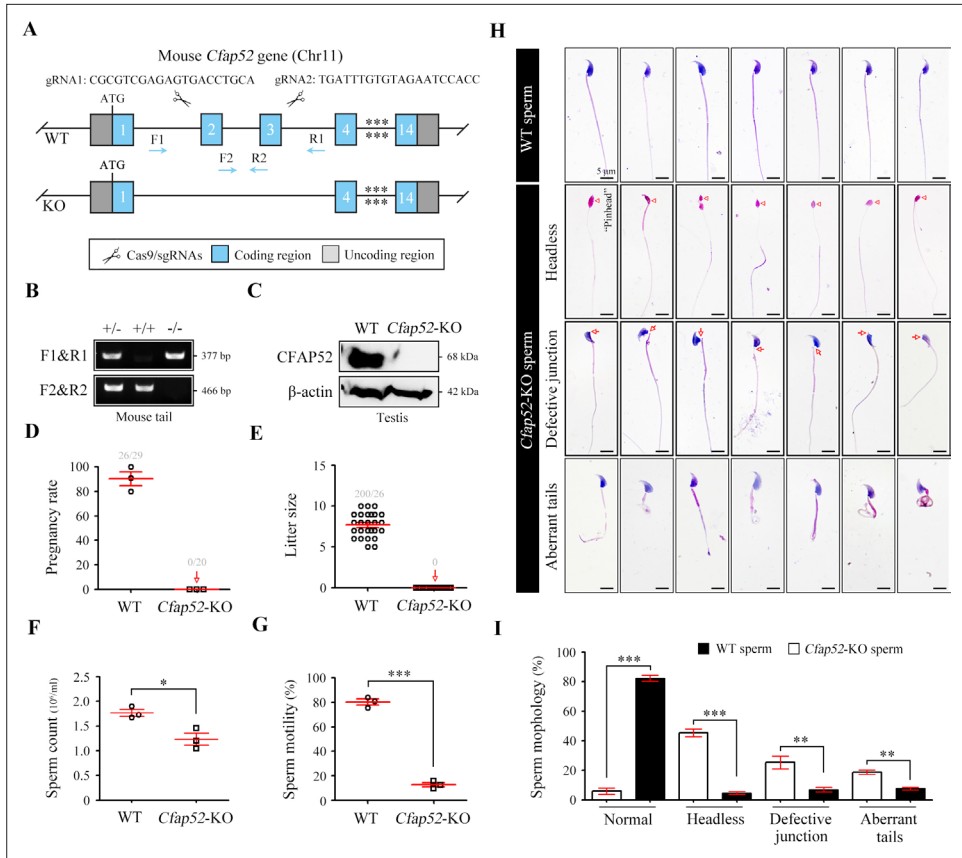

**Figure 4.** *Cfap52* deficiency in mice leads to male sterility, with spermatozoa showing defective head-tail connections and flagella. (**A**) Schematic illustration of the targeting strategy for generating *Cfap52*-KO mice by using CRISPR/Cas9 technology. A detailed procedure is described in the Materials and methods. (**B**) Representative results of PCR-based genotyping using mouse tail DNA. (**C**) Immunoblotting of CFAP52 was performed in the testis protein lysates of WT mice and *Cfap52*-KO mice. β-actin served as a loading control. Western blotting experiments were repeated three times with similar results. (**D, E**) Fertility assessment experiments were performed on three adult *Cfap52*-KO male mice and three WT male littermates for 2 months. Of the 20 female mice mated with *Cfap52*-KO male mice, no pregnancy was observed. (**F, G**) Sperm counts were counted with a fertility counting chamber under a light microscope, and total motility was assessed by a computer-assisted sperm analysis (CASA) system. Data are presented as the mean ± SEM (n=3 each group), Student's *t* test, *p<0.05; ***p<0.001. (**H**) Morphological analyses of spermatozoa in WT mice and *Cfap52*-KO mice by Papanicolaou staining. First row, normal morphology of spermatozoa from WT mice; second row, headless spermatozoa in *Cfap52*-KO mice; third row, spermatozoa with defective head-tail connection in *Cfap52*-KO mice; fourth row, short-tailed spermatozoa in *Cfap52*-KO mice. The arrowheads indicate the 'pinhead', and the arrows indicate the defective head-tail connection. Scale bars, 5 µm. (**I**) Percentage of spermatozoa with normal morphology and each type of defect in WT mice and *Cfap52*-KO mice. At least 100 spermatozoa were counted for each mouse. Data are presented as the mean ± SEM (n=3 each group), Student's *t* test, **p<0.01; ***p<0.001.

The online version of this article includes the following source data and figure supplement(s) for figure 4:

**Source data 1.** Primers for Cfap52-KO mouse genotyping.

**Source data 2.** Original blots of *Figure 4C*.

**Figure supplement 1.** Animal report of generation of *Cfap52*-KO mice.

**Figure supplement 2.** *Cfap52*-KO mice develop hydrocephalus.

**Figure supplement 3.** Detailed analysis of infertility in *Cfap52*-KO male and female mice.

**Figure supplement 4.** Periodic acid-Schiff staining of WT and *Cfap52*-KO testis sections.

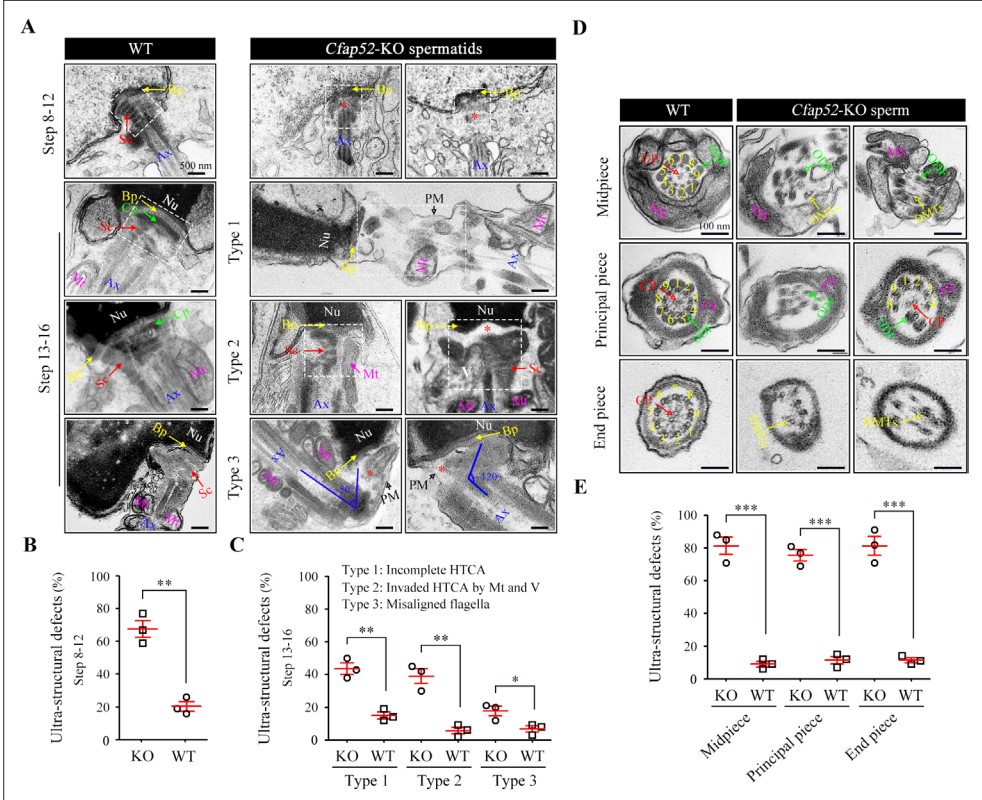

**Figure 5.** Impaired development of the connecting piece and the axoneme structure of spermatids/spermatozoa in *Cfap52*-KO mice. (**A**) Longitudinal sections of elongating (steps 10–12) and elongated (steps 13–16) spermatids in WT mice and *Cfap52*-KO mice, showing the structure of the connecting piece. Sc, segmented columns; Cp, capitulum; Bp, basal plate; Mt, mitochondria; Ax, axoneme. V, vacuoles; Nu, nucleus: PM, plasma membrane. Scale bars, 500 nm. (**B, C**) The percentage of spermatids with ultrastructural defects in the connecting piece of elongating spermatids and elongated spermatids from WT mice and *Cfap52*-KO mice. At least 30 spermatids were counted for each mouse. Data are presented as the mean ± SEM (n=3 each group), Student's *t* test, *p<0.05, **p<0.01. HTCA, head-tail coupling apparatus. (**D**) Cross-sections showing the ultrastructure of the midpiece, principal piece, and end piece of spermatozoa from WT mice and *Cfap52*-KO mice. CP, central pair; DMT, doublet microtubule; ODF, outer dense fibres; MS; mitochondrial sheath; FS, fibrous sheath. Scale bars, 100 nm. (**E**) The ratio of ultrastructural defects of the flagellar axoneme in the midpiece, principal piece, and end piece of spermatozoa from WT mice and *Cfap52*-KO mice. At least 30 spermatozoa were counted for each mouse. Data are presented as the mean ± SEM (n=3 each group), Student's *t* test, ***p<0.001.

## Impaired development of the sperm head-tail connecting piece and axoneme in the absence of CFAP52

Acephalic spermatozoa and abnormal head-tail connecting spermatozoa account for approximately two-thirds of total spermatozoa in the *Cfap52*-KO mice, suggesting that ensuring normal development of the connecting piece may be the major physiological effect of CFAP52. To identify the underlying structural defects, we examined the ultrastructure of spermatids in *Cfap52*-KO mice and WT mice using TEM technology (*Figure 5A*). In spermatids of WT mice, the connecting piece was consisted of well-defined Sc and tightly connected Cp-Bp structures (*Figure 5A*, left). In contrast, the connecting piece showed developmental defects in the spermatids of *Cfap52*-KO mice (*Figure 5A*, right). In details, an enlarged gap between the nuclear envelope and axoneme (type 1, the second row) as well as ectopically localized mitochondria and vacuoles in the connecting piece (type 2, the third row) were often identified in *Cfap52*-KO spermatids, indicating hypoplasia and/or instability of the connecting piece. Due to a deficiency of connecting piece formation, the developing flagella was often misaligned with the nuclei (type 3, the fourth row). Statistical analysis revealed that the percentage of spermatids (step 8~12) with ultra-structural defects of connecting piece in *Cfap52*-KO mice was significantly greater than that in WT mice (*Figure 5B*). Compared with those of the WT group, the ratios of spermatids

(step 13~16) with incomplete HTCA (type 1), invaded HTCA by mitochondria and vacuoles (type 2) or misaligned flagella (type 3) were all significantly increased in *Cfap52*-KO mice (*Figure 5C*).

In addition to a failure of connecting piece formation, CFAP52 deficiency led to seriously defects in flagellar development (*Figure 5D*). In the mid-piece, WT spermatozoa have a well-defined MS, ODFs and the inside typical '9+2' axoneme, whereas spermatozoa of *Cfap52*-KO mice displayed a disordered axoneme and accessory structures (*Figure 5D*, the first row). In the principle piece, axoneme and ODFs were partially lost in spermatozoa of *Cfap52*-KO mice (*Figure 5D*, the second row). The CP and DMTs were also absent and disordered in the end-piece of spermatozoa from *Cfap52*-KO mice (*Figure 5D*, the third row). Statistical analysis revealed that the ratios of spermatozoa with ultrastructural defects in the flagellar axoneme at the midpiece, principal piece, and end piece of *Cfap52*-KO mice were all significantly greater than those ratios in WT mice (*Figure 5E*).

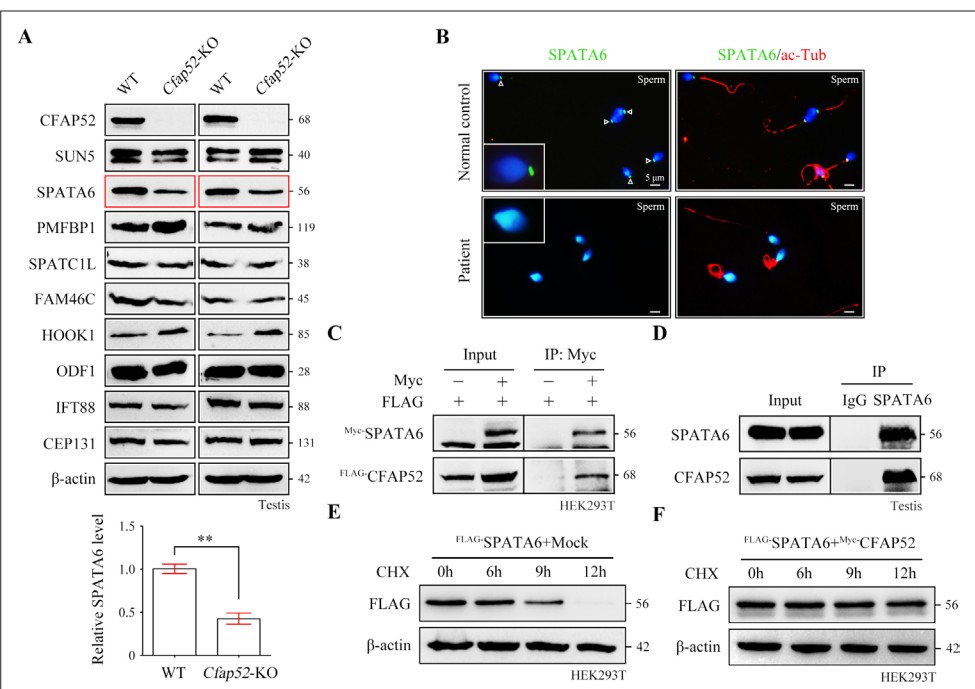

**Figure 6.** CFAP52 interacts with SPATA6 and regulates its expression. (**A**) Expression of nine ASS-associated proteins was analysed by western blotting in testis lysates from WT mice and *Cfap52*-KO mice. β-actin served as a loading control. Western blotting experiments were repeated three times with similar results. ASS, acephalic spermatozoa syndrome. Bar graph representing band intensities of SPATA6 blots, and data represent the mean ± SEM of three biological replicates. Student's *t* test, **p<0.01. (**B**) Coimmunofluorescence staining of SPATA6 (green) and ac-Tub (red) in spermatozoa from the *CFAP52*-mutant patient and a normal control. Scale bars, 5 μm. The staining experiments were repeated three times with similar results. (**C**) A coimmunoprecipitation assay showed that FLAG-tagged CFAP52 could be immunoprecipitated with Myc-tagged SPATA6 in HEK293T cell extracts. (**D**) An interaction between endogenous SPATA6 and CFAP52 was identified in mouse testis lysates. Rabbit IgG served as the negative control. Co-IP experiments were repeated three times with similar results. (**E, F**) The degradation of FLAG-tagged SPATA6 in HEK293T cells with or without Myc-tagged CFAP52. Protein samples were harvested at the indicated times after treatment with 100 μg/ml cycloheximide (CHX) to block new protein synthesis. β-actin served as a loading control. Western blotting experiments were repeated three times with similar results.

The online version of this article includes the following source data and figure supplement(s) for figure 6:

**Source data 1.** Primers for plasmid construction.

**Source data 2.** Original blots for *Figure 6A, C–F*.

**Figure supplement 1.** SUN5 staining in spermatozoa from the *CFAP52*-mutant patient and a healthy control.

## SPATA6 signals are undetectable in spermatozoa from *Cfap52*-KO mice and *CFAP52*-mutant man

To determine the molecular basis of the connecting piece defects in *Cfap52*-KO mice, we performed protein expression analysis of all known ASS-associated factors, such as SUN5, SPATA6, PMFBP1, SPATC1L, FAM46C, HOOK1, ODF1, IFT88, and CEP131, in the testis lysates of WT mice and *Cfap52*-KO mice. The expression of most ASS-associated factors was unaltered except for SPATA6 and HOOK1 (*Figure 6A*). SPATA6 expression was significantly attenuated in the testis protein lysates of *Cfap52*-KO mice compared with WT mice (*Figure 6A*). SPATA6 encodes a protein required for the formation of Sc and Cp (two major structures of the sperm HTCA), and inactivation of SPATA6 in mice leads to acephalic spermatozoa and male sterility (*Yuan et al., 2015*). The presence of SPATA6 was further examined in spermatozoa from a *CFAP52*-mutant patient and a fertile control man. SPATA6 signals were barely detectable in the spermatozoa of the *CFAP52*-mutant patient (*Figure 6B*). As a positive control, SUN5 staining at the connecting piece was not lost in spermatozoa from the *CFAP52*-mutant patient (*Figure 6—figure supplement 1*).

Both CFAP52 and SPATA6 was localized to the manchette of mouse elongating spermatids (*Tapia Contreras and Hoyer-Fender, 2020*; *Yuan et al., 2015*). The interaction between CFAP52 and SPATA6 was examined by co-immunoprecipitation (co-IP). FLAG-tagged CFAP52 could be immunoprecipitated with Myc-tagged SPATA6 in HEK293T cells (*Figure 6C*). The SPATA6-CFAP52 interaction was further confirmed by an endogenous co-IP experiment using testis proteins (*Figure 6D*). To determine whether CFAP52 regulates its interacting partner SPATA6 through the mechanism of protein stability, we treated HEK293T cells overexpressing SPATA6 with or without CFAP52 with cycloheximide for up to 12 hr to block protein synthesis. The degradation rate of SPATA6 was obviously slowed by the presence of the cofactor CFAP52 (*Figure 6E and F*). Taken together, these data suggest that CFAP52 is critical for the expression of its interacting partner SPATA6, which may be the mechanism underlying the ASS phenotype in CFAP52-deficient mice and humans.

## Dysexpression of axonemal proteins in spermatozoa from *Cfap52*-KO mice and the *CFAP52*-mutant man

CFAP52 contains WD repeat domains, which act as a protein interaction scaffolds in multiprotein complexes (*Li and Roberts, 2001*). Accordingly, we hypothesized that CFAP52 may interact with multiple axonemal components and regulate their expressions. CFAP52 has been reported to interact with axonemal components, including MIP (CFAP45), outer dynein arms (DNAI1 and DNAH11), and dynein regulatory complex (DRC10) (*Dougherty et al., 2020*). Given that available CFAP52 antibodies were not suitable for endogenous IP experiments, IP-mass spectrometry experiments could not be performed. Using co-IP assays in HEK293T cells, we identified novel axonemal interactors of CFAP52, including components of MIPs (CFAP20 and PACRG), radial spokes (RSPH1 and RSPH3), and outer dynein arms and their docking complexes (DNAI2, ODAD1, and ODAD3; *Figure 7—figure supplement 1*). CFAP52 did not interact with components of inner dynein arm (DNALI1), dynein regulatory complexes (DRC2 and DRC4), or central apparatus (SPAG6; *Figure 7—figure supplement 1*).

By western blotting, we found that the expression of components of MIPs (CFAP45 and ENKUR) and radial spokes (RSPH3 and RSPH9) was significantly reduced in sperm protein lysates of *Cfap52*-KO mice compared with WT mice (*Figure 7A*). The protein levels of components of outer dynein arms (DNAI1 and DNAI2), inner dynein arms (DNAH7 and DNALI1), and dynein regulatory complexes (DRC2 and DRC4) were not altered in sperm samples of *Cfap52*-KO mice (*Figure 7A*). By immunofluorescence staining, we further showed that the expression signals of CFAP45 and RSPH9 were almost absent in the sperm flagella from the *CFAP52*-mutant patient (*Figure 7B*). As a control, the expression of DNAI1 and DRC3 was not lost in the patient's sperm flagella (*Figure 7—figure supplement 2*).

## Treatment of *Cfap52*-KO mice and the *CFAP52*-mutant patient by ICSI

ICSI cycles were conducted using the spermatozoa from the *CFAP52*-mutant patient, and informed consent was obtained for the ICSI procedure. The patient's wife, who had a regular menstrual cycle and normal endocrine indices, underwent an antagonist protocol. Six follicles with a diameter of more than 14 mm and four follicles with a diameter of more than 18 mm were observed on the trigger day. We totally retrieved 31 oocytes. Then, 21 mature oocytes were microinjected, and 20 oocytes were ultimately fertilized (two-pronuclear/injected oocytes = 95.2%). All 20 embryos reached the cleavage

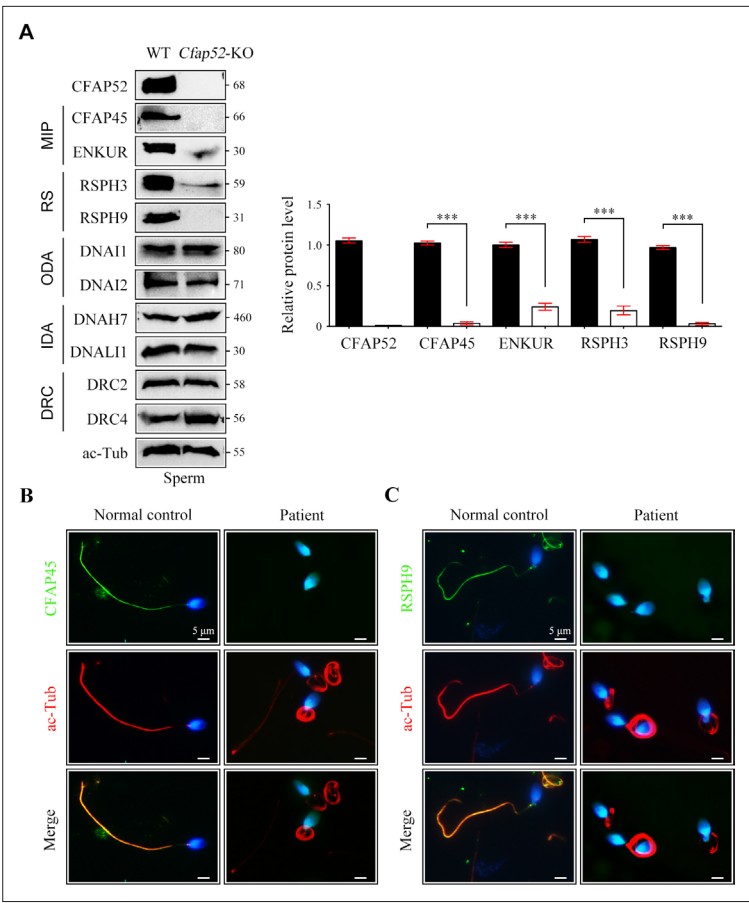

**Figure 7.** Reduced expression of components of MIP and RS in spermatozoa of *Cfap52*-KO mice. (**A**) Immunoblots of components of MIP, RS, ODA, IDA, DRC in the sperm protein lysates of WT mice and *Cfap52*-KO mice. Acetylated-tubulin (ac-Tub) served as a loading control. Grey values of bands were analysed by ImageJ software. The relative protein levels of CFAP45, ENKUR, RSPH3, and RSPH9 in sperm samples from WT mice and *Cfap52*-KO mice. Data represent the mean ± SEM of three biological replicates, and band intensities were normalized to ac-tub. Student's *t* test was performed, ***p<0.001. (**B, C**) Immunofluorescence staining of CFAP45 or RSPH9 (green) together with axonemal acetyl-tubulin (red) in spermatozoa from the *Cfap52*-mutant patient and a healthy control. The nuclei were counterstained with DAPI. Scale bars, 5 µm. The staining experiments were repeated three times with similar results.

The online version of this article includes the following source data and figure supplement(s) for figure 7:

**Source data 1.** Antibodies used in this study.

**Source data 2.** Original blots for *Figure 7A*.

**Figure supplement 1.** Axonemal interactors of CFAP52 as revealed by a coimmunoprecipitation assay.

**Figure supplement 1—source data 1.** Original blots for *Figure 7—figure supplement 1*.

**Figure supplement 2.** Expression of DNAI1 and DRC3 in spermatozoa of the *CFAP52*-mutant patient.

**Figure supplement 3.** ICSI outcomes using spermatozoa from *Cfap52*-KO mice.

---

stage and further reached the available D3 stage. After extension of the culture, three 4AA blastocysts, two 4AB blastocysts, five 4BC blastocysts, three 4BB blastocysts, and one 3BB blastocysts were obtained. Finally, the couple gave birth to a healthy newborn after one 4AA blastocyst was transplanted (*Table 3*). The results of ICSI for *Cfap52*-KO male mice was consistent with the observations in humans. Two-cell embryos and blastocyst were successfully obtained upon ICSI using the spermatozoa from *Cfap52*-KO mice, showing a similar efficiency with the control group (*Figure 7—figure supplement 3*). Overall, our data suggest that ICSI could serve as a promising treatment for infertile men with *CFAP52* variants.

**Table 3.** Clinical features of the patient's spouse with ICSI treatment.

|  |  | Spouse of the patient |
|---|---|---|
| Age(y) |  | 31 |
| Length of primary infertility history (y) |  | 2 |
| BMI |  | 20.5 |
| Basal hormones | FSH (IU/l) | 7.1 |
|  | LH (IU/l) | 4.7 |
|  | E2 (pg/ml) | 43.8 |
|  | Progesterone (ng/ml) | 0.26 |
| Cycle 1 | Protocol | Antagonist |
|  | E2 level on the trigger day (pg/ml) | 8101 |
|  | No. of follicles ≥14 mm on the trigger day | 6 |
|  | No. of follicles ≥18 mm on the trigger day | 4 |
|  | No. of oocytes retrieved | 31 |
| ICSI progress | Oocytes injected | 21 |
|  | Fertilization rate (%) | 95.2 (20/21) |
|  | Cleavage rate (%) | 100 (20/20) |
|  | Available D3 embryos | 20 |
|  | Blastocyst formation rate (%) | 70 (14/20) |
|  | 4AA | 3 |
|  | 4AB | 2 |
|  | 4BC | 5 |
|  | 4BB | 3 |
|  | 3BB | 1 |

## Discussion

CFAP52 is a member of the large WD40-repeat protein family and shows conserved expression in the cilia and flagella of eukaryotes. Model organism studies in *Chlamydomonas* and zebrafish have suggested that FAP52/cfap52 (two homologues of human CFAP52) play essential roles in cilia development. FAP52 and FAP45 are MIPs in *Chlamydomonas*, and lack of these proteins leads to less stable B-tubules (*Owa et al., 2019*). Loss of both FAP52 and FAP20 causes detachment of the B-tubule from the A-tubule and shortened flagella (*Owa et al., 2019*). The effect of FAP52 on axoneme structure seems to be dependent on other co-factors because loss of FAP52 alone does not generate such an obvious phenomenon. In zebrafish, *cfap52* knockdown by antisense morpholino injection leads to hydrocephalus formation (*Hirschner et al., 2007*). Several clinical investigations further highlight the potential contribution of CFAP52 on human cilia development. A homozygous deleterious deletion in *CFAP52* is identified in patients with situs anomalies (laterality disorder) (*TaShma et al., 2015*). Homozygous mutations in *CFAP52* were further reported among four individuals of unrelated pedigrees with situs inversus totalis and mild chronic upper respiratory symptoms (*Dougherty et al., 2020*). However, the physiological functions of CFAP52 in mammals and the association of *CFAP52* mutations with sperm flagella disorders in humans remain unknown.

In this study, pathogenic mutations in *CFAP52* were identified by WES in a Chinese infertile man with asthenoteratospermia (a mixed phenotype of ASS and MMAF). Notably, no abnormalities in motile cilia (e.g. hydrocephalus, situs inversus, and chronic upper respiratory symptoms) were observed in the patient. *Cfap52*-KO mice were further generated and exhibited male infertility with a mixed phenotype of ASS and MMAF, indicating that CFAP52 regulates the development of the HTCA

and flagella. Moreover, *Cfap52*-KO mice showed hydrocephalus but no other ciliopathies, including situs inversus and abnormalities of tracheal cilia. Although CFAP52 is widely expressed in a subset of cilia/flagella structures, it may be specifically required for some tissues and organs. The *CFAP52*-mutant patient has only fertility problem, while *Cfap52*-KO mice display hydrocephalus and male infertility. We suggest that no obvious role or functional redundancy of CFAP52 in the development of ependymal cilia in humans. Another explanation is that mice are more sensitive to CFAP52 deficiency or develop hydrocephalus more easily than humans.

A previous study suggested that CFAP52, similar to its interacting partner CFAP45, may regulate axonemal adenine nucleotide (ADP, AMP and ATP) homeostasis (*Dougherty et al., 2020*). However, we showed here that the phenotype of *Cfap52*-KO mice was different from that of *Cfap45*-KO mice, which produce immobile sperm with normal flagellar length. In contrast, the sperm of *Cfap52*-KO mice showed defective head-tail connection and deformed flagella. Moreover, the *CFAP52*-mutant patient exhibited abnormalities in the development of HTCA and flagellar axoneme, which is consistent with the phenotype of *Cfap52*-KO mice.

CFAP52 is a centrosome/basal body protein and localizes to the manchette of mouse spermatids (*Tapia Contreras and Hoyer-Fender, 2020*). The manchette serves as a 'conveyer belt' mediating the transportation and processing of proteins required for the HTCA and flagellar assembly (*Kierszenbaum et al., 2011*). A candidate-based approach was applied to examine the regulation of known ASS-associated proteins by CFAP52. The expression of most ASS-associated proteins were not altered after the CFAP52 deletion. Only SPATA6 expression was significantly reduced in Cfap52-KO testes as compared with that in WT testes. Although no mutations in SPATA6 have been identified in ASS patients, SPATA6 is a well-known structural protein of HTCA and is required for the formation of Sc and Cp via its interaction with myosin subunits (e.g. MYL6) in mice (*Yuan et al., 2015*). We further showed that CFAP52 interacted with the middle region of SPATA6 via its first eight WD40 domains to regulate the stability of SPATA6. The mechanism underlying the regulation of SPATA6 by CFAP52 requires further investigation and is not the main focus of our current study.

Unlike a specific regulation of the stability of B-tubules by FAP52 in *Chlamydomonas* (*Owa et al., 2019*), *Cfap52*-KO mice and *CFAP52*-mutant patient showed a serious disorder of the axoneme and its accessory structures. Compared with the MIP-specific expression of FAP52 in *Chlamydomonas*, CFAP52 seems to interact with a broad range of axonemal proteins, including MIP (CFAP45), ODAs (DNAI1 and DNAH11), and DRC (DRC10) in mammals (*Dougherty et al., 2020*). Herein, we extended the axonemal interactors of CFAP52 in mammals and suggest that an attenuated expression of components of MIPs (CFAP45 and ENKUR) and radial spokes (RSPH3 and RSPH9) partially contributed to the severe MMAF phenotype of *Cfap52*-KO sperm. The mechanism underlying the regulation of flagella development by CFAP52 requires further investigation.

During our manuscript is under preparation, an independent group also generated the *Cfap52*-KO mice and explored their phenotype (*Wu et al., 2023*). Male infertility with reduced sperm motility is the common phenotype of two *Cfap52*-KO mouse models; however, an in-depth analysis of the phenotype and underlying mechanism is distinct. In the JBC paper, *Cfap52*-KO mice display a disorganized junction of midpiece and principal piece; however, the '9+2' axonemal structure is normal. This phenotype may be somewhat not consistent with previous observations: (i) FAP52 is an axoneme structural protein (termed microtubule inner protein, MIP) in *Chlamydomonas*, and a lack of FAP52 leads to less stable B-tubules (*Owa et al., 2019*); (ii) CFAP52 is distributed along the sperm flagella rather than in a specific region between the midpiece and principal piece (*Tapia Contreras and Hoyer-Fender, 2020*). Our study supports the notion that CFAP52 is an axonemal protein of sperm flagella because loss of CFAP52 led to a disruption of '9+2' axonemal structures. In both studies, the C57BL/6 J strain of mice was used, and the absence of CFAP52 was confirmed in the testis protein lysates by western blotting. Only the targeting strategies are different between these two studies. In the JBC paper, exons 2~6 of mouse *Cfap52* were selected as the targeting region. In our study, exons 2 and 3 of mouse *Cfap52* were targeted, and the gRNA sequences and technical monitoring of *Cfap52* gene targeting are provided. Given that complete loss of CFAP52 protein expression was confirmed in both studies, the different targeting strategy may not be the reason underlying the phenotypic difference between the two *Cfap52*-KO mouse models.

Before our study, no direct evidence indicated the relevance of *CFAP52* mutations to human male infertility. Herein, compound heterozygous variants of *CFAP52* were identified in a Chinese infertile

man with asthenoteratospermia. The absence of another independent pedigree is a limitation of this study. Identification of pathogenic *CFAP52* variants in a larger cohort of male infertile men with ASS and/or MMAF is needed to support the conclusion that *CFAP52* is a solid asthenoteratospermia-associated gene. Moreover, healthy offspring could be obtained via ICSI using spermatozoa from *Cfap52*-KO mice and *CFAP52*-mutant patients, showing the clinical value of the research. Hence, we suggest that ICSI could serve as a promising treatment for infertile men harbouring pathogenic *CFAP52* variants. More cases still need to be investigated to clarify the consequence of *CFAP52* mutations on ICSI outcomes.

In conclusion, this study showed that the disruption in *CFAP52* causes male infertility in humans and mice. Our findings help improve the understanding of the formation of head-tail connecting pieces and flagella as well as the genetic causes of male infertility. *CFAP52* is expected to be developed as a novel target for the genetic diagnosis of asthenoteratospermia. Our study also suggests that ICSI is a promising technology for *CFAP52*-mutant infertile men.

## Materials and methods

### Study participants

An infertile man and his parents were recruited at the West China Second Hospital of Sichuan University. The human study was performed in accordance with the principles of the Declaration of Helsinki. Each subject signed informed consent. This study was approved by the Ethical Review Board of West China Second University Hospital, Sichuan University (Ethic Review [Scientific Research] to Shen Ying).

### WES and Sanger sequencing

Peripheral blood of the subjects was used to extract genomic DNA via a QIAamp DNA Blood Mini Kit (QIAGEN, USA). The Agilent SureSelect Human All Exon V6 Kit (Agilent Technologies, USA) was applied for exon capture, and the Illumina HiSeq X system was utilized to perform sequencing. Reads were mapped to the human genome reference (GRCh37/hg19) by Burrows Wheeler Aligner (BWA) software. ANNOVAR software was used for functional annotation. The Genome Analysis Toolkit was employed to identify and quality filter the variants. Sanger sequencing was applied to verify the variants detected by WES in the patient and his parents. The primers for Sanger sequencing are shown in *Figure 3—source data 1*.

### Real-time PCR (RT-PCR)

TRIzol (Invitrogen, USA) and a RevertAid First-Strand cDNA Synthesis Kit (Thermo Fisher) were used to extract total RNA and to convert the total RNA to cDNA, respectively. Next, real-time PCR was performed with SYBR Premix Ex Taq II (TaKaRa, China) on an iCycler RT−PCR Detection System (Bio-Rad Laboratories, USA). ΔΔCT method was used for data analysis. The *Gapdh* gene was used as an internal control. The primers for RT−PCR are shown in *Figure 3—source data 1*.

### Minigene assay

A mini-gene splicing assay was used to examine the effect of the variant in *CFAP52* (c.203G>T) on splicing. In brief, intron 1, exon 2, and intron 2 of *CFAP52* sequences were amplified from genomic DNA of the healthy control and the patient. Amplified fragments were cloned into the minigene vector pSPL3. Splicing patterns of the transcripts were determined after HEK293T cell transfection by RT−PCR, gel electrophoresis and Sanger sequencing. The primers are provided in *Figure 3—source data 1*.

### Knockout mice and animal studies

Animal experiments were approved by the Animal Care and Use Committee of the College of Life Sciences, Beijing Normal University (CLS-AWEC-B-2023–001). Exons 2~3 of the *Cfap52-201* (ENSMUST00000021287.11) transcript were selected as the target region. The CRISPR/Cas9 technology to generate knockout mice were described in our previous studies (*Zhang et al., 2022c*; *Zhang et al., 2022d*). The sgRNAs are provided in *Figure 4—figure supplement 1* and primers for genotyping are listed in *Figure 4—source data 1*. The detailed procedure of fertility testing and

intracytoplasmic sperm injection (ICSI) was performed as previously described (*Jin et al., 2023*; *Zhang et al., 2022a*).

## Expression plasmids and transient transfection

FLAG- and/or Myc-tagged pCMV vectors were used to construct expression plasmids. Mouse *CFAP52* cDNA was produced by Sangon Biotech (Shanghai, China) and mouse testis cDNA template was used to amplify other genes by PCR. The primers for PCR amplification are listed in *Figure 6—source data 1*. Lipofectamine 3000 transfection reagent (Invitrogen, USA) was used to transfect HEK293T cells (from ATCC) and cycloheximide (MedChemExpress, Shanghai, China) was added to culture to block protein synthesis. HEK293T line was obtained from ATCC (American Type Culture Collection) and authenticated using STR profiling test by Shanghai Biowing Applied Biotechnology Co., Ltd.

## Sperm count, motility and morphology

Sperm counts were assessed using a Fertility Counting Chamber (Makler, Israel). A computer-assisted sperm analysis (CASA) system (SAS Medical, China) was used to determine sperm motility. Semen smears were stained with Papanicolaou solution (Solarbio, China) and photographed using a light microscope.

## Immunofluorescence

Sperm slides were permeabilized and blocked with 5% goat serum (Beyotime, China) for 45 min. Primary antibodies were treated overnight at 4 °C. After three washes with PBS, Alexa Fluor 484- or 555-labeled donkey anti-rabbit IgG (Beyotime, China) was incubated for 45 min. The nuclei were counterstained with DAPI dye (Beyotime, China). The antibodies used in immunofluorescence staining are provided in *Figure 7—source data 1*.

## Electron microscopy

Samples were fixed with 2.5% glutaraldehyde (Zhongjingkeyi Technology, China) in 0.1 M phosphate buffer (PB) (pH 7.4). The detailed procedure of transmission electron microscopy (TEM) and scanning electron microscopy (TEM) was described in our previous studies (*Jin et al., 2023*; *Zhang et al., 2022b*; *Zhang et al., 2022c*). For TEM, specimens were photography with a Tecnai G2 Spirit 120 kV (FEI) electron microscope; for SEM, specimens were viewed with a JSM-IT300 scanning electron microscope (JEOL, Japan).

## Intracytoplasmic sperm injection

C57BL/6 J female mice were superovulated by administration with 10 IU PMSG combined with 10 IU hCG (48 hr later). Oocytes were obtained from the ampulla of the uterine tube at 14 hr after hCG injection. Mouse sperm heads were separated from sperm tails and injected into mouse oocytes through using a Piezo driven pipette (PrimeTech, Japan). The injected oocytes were cultured in KSOM medium at 37 °C under 5% $CO_2$. All reagents were purchased from Nanjing Aibei Biotechnology.

## Coimmunoprecipitation (co-IP) and western blotting

The detailed procedure of co-IP and western blotting was described in our previous studies (*Jin et al., 2023*; *Zhang et al., 2022c*; *Zhang et al., 2022d*). Pierce IP Lysis Buffer and Pierce Protein A/G-conjugated Agarose (Thermo Fisher, USA) were used in co-IP experiment. For in vitro co-IP, 2 µg anti-Myc antibody or anti-FLAG antibody was added to the protein lysates of transfected HEK293T cells. For endogenous co-IP, 2 µg anti-SPATA6 antibody or 2 µg rabbit IgG was added to the protein lysates of mouse testes. Antibodies used in co-IP and WB are listed in *Figure 7—source data 1*.

## Data analysis and statistics

GraphPad Prism version 5.01 (GraphPad Software, RRID:SCR_002798) was used for data processing. Data are presented as the mean ± SEM. Unpaired, two-tailed Student's *t* test was used for the statistical analyses; *$p<0.05$, **$p<0.01$ and ***$p<0.001$.

# Acknowledgements

We appreciate the patients and their family members for their support during this research study. We would like to thank Jin Liu from Experimental Technology Center for Life Sciences, Beijing Normal University for the assistance of cell culture. We also acknowledge Xi-Xia Li and Shuang-Zhong Lv from the Center for Biological Imaging (CBI), Institute of Biophysics, Chinese Academy of Sciences for their help in making TEM samples.

## Additional information

### Funding

| Funder | Grant reference number | Author |
|---|---|---|
| National Natural Science Foundation of China | 32370905 | Su-Ren Chen |
| National Key Research and Development Program of China | 2019YFA0802101 | Su-Ren Chen |
| Open Fund of Key Laboratory of Cell Proliferation and Regulation Biology, Ministry of Education | | Su-Ren Chen |

The funders had no role in study design, data collection and interpretation, or the decision to submit the work for publication.

### Author contributions

Hui-Juan Jin, Tiechao Ruan, Validation, Investigation, Methodology; Siyu Dai, Validation, Methodology; Xin-Yan Geng, Software, Validation, Methodology; Yihong Yang, Formal analysis, Supervision, Validation, Investigation; Ying Shen, Software, Formal analysis, Supervision, Validation, Investigation; Su-Ren Chen, Formal analysis, Supervision, Funding acquisition, Validation, Investigation, Writing – original draft, Project administration, Writing – review and editing

### Author ORCIDs

Tiechao Ruan 
Su-Ren Chen 

### Ethics

The human study was performed in accordance with the principles of the Declaration of Helsinki. Each subject signed informed consent. This study was approved by the Ethical Review Board of West China Second University Hospital, Sichuan University (Ethic Review [Scientific Research] to Shen Ying).

Animal experiments were approved by the Animal Care and Use Committee of the College of Life Sciences, Beijing Normal University (CLS-AWEC-B-2023-001). All surgery was performed under sodium pentobarbital anesthesia, and every effort was made to minimize suffering.

Reviewer #1 (Public Review): https://doi.org/10.7554/eLife.92769.2.sa1
Reviewer #2 (Public Review): https://doi.org/10.7554/eLife.92769.2.sa2
Reviewer #3 (Public Review): https://doi.org/10.7554/eLife.92769.2.sa3
Author Response https://doi.org/10.7554/eLife.92769.2.sa4

## Additional files

### Supplementary files
• MDAR checklist

## Data availability

All data generated or analysed during this study are included in the manuscript and supporting files. Information of primers, sequences, antibodies, uncropped gels and blots are available at corresponding source data files.

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
