## [Editor Report · eLife assessment]

This study provides **useful** information on the function of a ciliary and flagellar-associated protein, CFAP52, in the assembly of sperm head-tail connecting apparatus (HTCA) and tail formation in humans and mice. The significance is to identify CFAP52 as a genetic factor for asthenoteratozoospermia with a mixed acephalic spermatozoa syndrome (ASS) and multiple morphological abnormalities of the sperm flagella (MMAF) phenotype. The strength of the study is that the experimental evidence using CFAP52 loss-of-function in mice is **solid** to support that CFAP52 is essential for sperm motility and male fertility by contributing to HTCA and 9+2 axoneme, corroborating the sperm phenotypes of human patients with compound heterozygous mutations in CFAP52.

---

## [Referee Report · Reviewer #1 (Public Review)]

In this manuscript, the authors identified compound heterozygous mutations in CFAP52 recessively cosegregating with male infertility status in a non-consanguineous family. The Cfap52-mutant patient exhibits a mixed acephalic spermatozoa syndrome (ASS) and multiple morphological abnormalities of the sperm flagella (MMAF) phenotype. The influence of mutations on CFAP52 protein function is well validated by in vitro cell experiments and immunofluorescence staining. Cfap52-KO mice are further constructed and perfectly resemble the Cfap52-mutant patient's infertile phenotype, also showing a mixed ASS and MMAF phenotype. The phenotype and underlying mechanisms of the disruption of sperm head-tail connection and flagella development are carefully analyzed by TEM, Western blotting, and immunofluorescence staining. The data presented revealed a prominent role for CFAP52 in sperm development, suggesting that CFAP52 is a novel diagnostic target for male infertility with defects of sperm head-tail connection and flagella development.

---

## [Referee Report · Reviewer #2 (Public Review)]

Summary:

The authors tried to identify the genetic factors for asthenoteratozoospermia. Using whole-exome sequencing, they analyzed a family with an infertile male and identified CFAP52 variants. They further knockout mouse Cfap52 gene and the homozygous mice phenocopied the patient. CFAP52 interacts with several other sperm proteins to maintain normal sperm morphology. Finally, CFAP52-associated male infertility in humans and mice could be overcome by using intracytoplasmic sperm injections (ICSI).

Strengths:

The major strength of this study is to identify genetic factors contributing to asthenoteratozoospermia, and to generate a mouse knockout model to validate the factor.

Weaknesses:

The authors did not use the OMICS to dissect the potential mechanisms. Instead, they took the advantage of direct co-IP experiment to fish the binding partners. They also did not discuss in detail why other motile cilia have different behavior.

---

## [Referee Report · Reviewer #3 (Public Review)]

Summary:

In this study, Jin et al. report the first evidence of CFAP52 mutations in human male infertility by identifying deleterious compound heterozygous mutations of CFAP52 in infertile human patients with acephalic and multiple morphological abnormalities in flagella (MMAF) phenotypes but without other abnormalities in motile cilia. They validated the pathogenicity of the mutations by an in vitro minigene assay and the absence of proteins in the patient's spermatozoa. Using a Cfap52 knockout mouse model they generated, the authors showed that the animals are hydrocephalic and the sperm have coupling defects, head decapitation, and axonemal structure disruption, supporting what was observed in human patients.

Strengths:

The major strengths of the study are the rigorous phenotypic and molecular analysis of normal and patient spermatozoa and the demonstration of infertility treatment by ICSI. The authors demonstrated the interaction between CFAP52 and SPATA6, a head-tail coupling regulator and structural protein, and showed that CFAP52 can interact with components of the microtubule inner protein (MIP), radial spoke, and outer dynein arm proteins.

Weaknesses:

The weakness of the study is some inconsistency in the localization of the CFAP52 protein in human spermatozoa in the figures and the lack of such localization information completely missing in mouse spermatozoa. Putting their findings in the context of the newly available structural information from the recent series of unambiguous and unequivocal identification of CFAP52 as an MIP in the B tubule will not only greatly benefit the interpretation of the study, but also resolve the inconsistent sperm phenotypes reported by an independent study. Since the mouse model is not designed to exactly recapitulate the human mutations but a complete knockout and the knockout mice show hydrocephaly phenotype as well, some of the claims of causality and ICSI as a treatment need to be tempered. Discussing the frequency of acephaly and MMAF in primary male infertility will be beneficial to justify CFAP52 as a practical diagnostic tool.

---

## [Author Response]

**Reviewer #1 (Public Review):**
In this manuscript, the authors identified compound heterozygous mutations in CFAP52 recessively cosegregating with male infertility status in a non-consanguineous family. The Cfap52-mutant patient exhibits a mixed acephalic spermatozoa syndrome (ASS) and multiple morphological abnormalities of the sperm flagella (MMAF) phenotype. The influence of mutations on CFAP52 protein function is well validated by in vitro cell experiments and immunofluorescence staining. Cfap52-KO mice are further constructed and perfectly resemble the Cfap52-mutant patient's infertile phenotype, also showing a mixed ASS and MMAF phenotype. The phenotype and underlying mechanisms of the disruption of sperm head-tail connection and flagella development are carefully analyzed by TEM, Western blotting, and immunofluorescence staining. The data presented revealed a prominent role for CFAP52 in sperm development, suggesting that CFAP52 is a novel diagnostic target for male infertility with defects of sperm head-tail connection and flagella development.

Thank you for your positive comments.

**Reviewer #2 (Public Review):**
Summary:The authors tried to identify the genetic factors for asthenoteratozoospermia. Using whole-exome sequencing, they analyzed a family with an infertile male and identified CFAP52 variants. They further knockout mouse Cfap52 gene and the homozygous mice phenocopied the patient. CFAP52 interacts with several other sperm proteins to maintain normal sperm morphology. Finally, CFAP52-associated male infertility in humans and mice could be overcome by using intracytoplasmic sperm injections (ICSI).Strengths:The major strength of this study is to identify genetic factors contributing to asthenoteratozoospermia, and to generate a mouse knockout model to validate the factor.

Thank you for your positive comments.

Weaknesses:The authors did not use the OMICS to dissect the potential mechanisms. Instead, they took the advantage of direct co-IP experiment to fish the binding partners. They also did not discuss in detail why other motile cilia have different behavior.

Dear reviewer, thank you for your comments and we tried to answer your two questions as follows.

In this study, we did not choose omics technologies to explore the binding partners for CFAP52 (e.g., IP-MS) and differentially expressed proteins after the loss of CFAP52 (e.g., proteomics). For IP-MS, we feel sorry that all available antibodies of CFAP52 could not be used to perform protein immunoprecipitation experiments. Another reason is that there are only dozens of proteins that have been reported to regulate the head-tail coupling apparatus (HTCA) of sperm. Accordingly, we used Western blotting to examine the expression of ten acephalic sperm syndrome (ASS)-associated proteins and found that only SPATA6 expression was significantly reduced in the testis protein lysates of Cfap52-KO mice (Fig. 6A). We further carefully examined the regulation of the stability of SPATA6 by its binding partner CFAP52 (Fig. 6 and Figure 6—figure supplement 2).

In addition to male infertility, Cfap52-KO mice suffered from hydrocephalus; the ependymal cilia was sparse under SEM observation and disrupted axonemal structures were identified by TEM analysis (Figure 4—figure supplement 2). However, no obvious abnormalities of tracheal cilia were identified by SEM and TEM analyses (Figure 4—figure supplement 2). Although flagella and motile cilia exhibit quite similar “9+2” axoneme structure, they have some their unique proteins and the requirement of some axonemal proteins may be different. For example, IQUB expression is detected in tissues other than the testis, such as the lung and brain; however, IQUB deletion only affects beating of sperm flagella but not respiratory cilia (Cell Rep, 2022). Cfap43-KO mice exhibited both sperm flagella disordor and early-onset hydrocephalus (Dev Biol, 2020), and CFAP206 is required for sperm motility, mucociliary clearance of the airways and brain development (Development, 2020).

**Reviewer #3 (Public Review):**
Summary:In this study, Jin et al. report the first evidence of CFAP52 mutations in human male infertility by identifying deleterious compound heterozygous mutations of CFAP52 in infertile human patients with acephalic and multiple morphological abnormalities in flagella (MMAF) phenotypes but without other abnormalities in motile cilia. They validated the pathogenicity of the mutations by an in vitro minigene assay and the absence of proteins in the patient's spermatozoa. Using a Cfap52 knockout mouse model they generated, the authors showed that the animals are hydrocephalic and the sperm have coupling defects, head decapitation, and axonemal structure disruption, supporting what was observed in human patients.Strengths:The major strengths of the study are the rigorous phenotypic and molecular analysis of normal and patient spermatozoa and the demonstration of infertility treatment by ICSI. The authors demonstrated the interaction between CFAP52 and SPATA6, a head-tail coupling regulator and structural protein, and showed that CFAP52 can interact with components of the microtubule inner protein (MIP), radial spoke, and outer dynein arm proteins.

Thank you for your positive comments.

Weaknesses:The weakness of the study is some inconsistency in the localization of the CFAP52 protein in human spermatozoa in the figures and the lack of such localization information completely missing in mouse spermatozoa. Putting their findings in the context of the newly available structural information from the recent series of unambiguous and unequivocal identification of CFAP52 as an MIP in the B tubule will not only greatly benefit the interpretation of the study, but also resolve the inconsistent sperm phenotypes reported by an independent study. Since the mouse model is not designed to exactly recapitulate the human mutations but a complete knockout and the knockout mice show hydrocephaly phenotype as well, some of the claims of causality and ICSI as a treatment need to be tempered. Discussing the frequency of acephaly and MMAF in primary male infertility will be beneficial to justify CFAP52 as a practical diagnostic tool.

Dear reviewer, thank you for your comments and we tried to answer your questions as follows.

By immunofluorescence staining, we showed that CFAP52 was localized at both HTCA and full-length flagella from the normal control; in contrast, CFAP52 signals were barely detected in the patient’s spermatozoa (Figure 3F). Given that CFAP52 staining did not occur in other figures, no inconsistency exists in the localization of the CFAP52 protein in human spermatozoa in the figures. We did not perform the CFAP52 staining in mouse spermatozoa; however, we have shown that CFAP52 protein was completely absent in the Cfap52-KO testes compared with the WT testes (Figure 4C).

We appreciate the reviewer’s suggestion to put our findings of CFAP52 in the context of the newly available axoneme architecture. Given that these cryo-EM studies focus on doublet microtubules (DMTs), a broader expression pattern of CFAP52 in cilia/flagella could not be excluded. In mammals, CFAP52 seems to interact with a broad range of axonemal proteins, including MIP (CFAP45), ODAs (DNAI1 and DNAH11), and DRC (DRC10) (Dougherty et al., 2020). We have mentioned that ‘a lack of FAP52 in Chlamydomonas causes an instability of microtubules and detachment of the B-tubule from the A-tubule and shortened flagella are observed in Chlamydomonas when both FAP52 and FAP20 are absent (Owa et al., 2019). Unlike a specific regulation of the stability of B-tubules by FAP52 in Chlamydomonas (Owa et al., 2019), Cfap52-KO mice and CFAP52-mutant patient showed a serious disorder of the axoneme and its accessory structures.’

Before our study, Cfap52-KO mice have not yet been generated. To explore the physiological roles of CFAP52, we decided to construct Cfap52-KO mice. During our manuscript is under preparation, an independent group also generated the Cfap52-KO mice and explored their phenotype (Wu et al., 2023). We quite agree with this reviewer that Cfap52-mutant mice will be exact models to recapitulate the human variants. Cfap52-mutant mice were not included in our current manuscript due to (i) the two identified variants were ‘nonsense’ variant and ‘frameshift’ variants, respectively, which are expected to damage the CFAP52 expression and function; (ii) the influence of two variants on CFAP52 protein function has been well validated by in vitro cell experiments and (iii) research funding is limited for us. The assisted reproductive technology (ART) outcomes were also reported for the CFAP52-mutant patient and Cfap52-KO mice, which will be potential useful for further clinical studies. However, it is not suggested to be over-interpreted because it is only a case study.

Quantitative analyses showed that the decapitated spermatozoa, abnormal head-tail connecting spermatozoa, and spermatozoa with deformed flagella accounted for approximately 40%, 25%, and 30% of the total spermatozoa in Cfap52-KO mice, respectively (Figure 4I). Regarding the CFAP52-mutant patient, the frequency of acephaly and MMAF were not counted and now we feel sorry that we don’t have enough samples (repeats) to perform quantitative analyses.